# Factors Affecting Users’ Continuous Usage in Online Health Communities: An Integrated Framework of SCT and TPB

**DOI:** 10.3390/healthcare11091238

**Published:** 2023-04-26

**Authors:** Zhuolin Cao, Jian Zheng, Renjing Liu

**Affiliations:** 1School of Management, Xi’an Jiaotong University, No. 28, Xianning West Road, Xi’an 710048, China; 2School of Pharmacy, Xi’an Medical College, Xi’an 710021, China

**Keywords:** online health community, continuous usage, SCT, TPB, structural equation

## Abstract

Background: Online health communities (OHCs) provide a new channel for users to obtain more health-related information and support, playing an important role in alleviating hospital congestion and uneven medical resource distribution, especially during the COVID-19 pandemic in China. An in-depth study of users’ continuous usage is of great value for the long-term development of OHCs. Objective: The purpose of this study is to explore the factors that influence users’ continuous usage in online health communities based on the theory of planned behavior (TPB) and social cognitive theory (SCT). Methods: Data from 480 users with experience in online health communities were collected through a questionnaire survey, and structural equations were applied to verify the model hypotheses empirically. Results: Self-efficacy and controllability have significant effects on users’ continuous intention; attitude has a significant relationship with continuous intention; social norms have a positive effect on continuous intention. Moreover, the relationship between continuous intention and behavior is positive. Self-efficacy and outcome expectations have significant positive associations with continuous usage. Finally, system quality, information quality, and social interaction ties have significant and positive relationships to continuous usage. Conclusion: To improve the level of user’s continuous usage, online health service providers can improve the quality of the community by organizing the website’s page layout, navigation menus, and site elements to ensure users quickly search and find what they want meanwhile try to change people’s cognition gradually, in addition, decision and policymakers should provide more favorable policies to stimulate and help provider in building and managing strategic plans for sustaining a thriving online community. A supportive climate in society through public service advertisements and others for the sake of OHCs is necessary. Limitations: (1) This study collected data through a cross-sectional survey. Thus, it lacked the process of capturing the changes in participants’ attitudes toward all variables. (2) The environmental factors in SCT theory need to be more comprehensive, containing online factors without offline factors. (3) The dates were obtained from China, which neglects the different cultural content.

## 1. Introduction

With the rapid growth of Internet technology and Web 2.0, virtual communities stepped and penetrated all aspects of citizens. As the representation of the virtual community, online health communities obtain rapid expansion and bring leverage to ordinary citizens, for they are informative, specific, flexible, and disinhibited [1]. What is more, the development of OHCs has brought social value, such as improving relations between doctors and patients and alleviating rural–urban health disparities [2]. Studies have shown that online health communities play an important role during infectious disease pandemics [3,4,5]. Especially the Chinese online health communities have provided more convenient professional medical consultation services and reduced the risk of patients’ infection during the coronavirus 2019 disease (COVID-19) pandemic. The OHCs are a convenient consultation mode during the epidemic prevention and control period and an essential part of future community development and users’ medical treatment. It has been widely observed within the normative literature that continuance participation is focal, and the social values and potential of online communities cannot be realized without users’ ongoing participation [6]. Thus, there is great realistic significance in clarifying the influencing factors of users’ continuous use for community development and users’ medical treatment.

Research on OHCs in China is still in its infancy [7]. Previous studies mainly focus on knowledge-sharing behavior [8,9,10], social communication [11,12,13,14], and social support behavior in the continuous usage of OHCs [15,16,17]. Less discussion of factors that affect the continuous usage of OHCs caused it difficult for community managers to grasp the critical influencing factor and maintain OHCs’ ongoing development. In this context, we establish a model of factors affecting users’ continuous usage of OHCs, which contains the theory of planned behavior and social cognition theory. They systematically explain the factors that influence individual behavior from the perspective of cognition and environments. The results confirm that both personal factors (self-efficiency and outcome expectation) and environmental factors (system quality, information quality, and social interaction ties) have positive effects on users’ intentions and behavior.

This research is organized as follows: The theoretical framework summarizes the main theories used in this research and puts forward the framework. Then, the research hypotheses are proposed. Then, the research methods used in this study are discussed. After this, we present our research results. We then discuss the main findings and propose theoretical contributions, practical implications, and limitations. Finally, we provide the research conclusion.

## 2. Literature Review

### 2.1. Online Health Communities and Continuous Usage

Online health communities are platforms that allow doctors to use existing medical resources to serve remote patients [18]. They also provide paths for patients to find information, maintain relationships, and obtain suggestions and support related to health. They broaden and diversify interaction channels between doctors and patients using Internet technology [19,20]. Moreover, The OHCs can enhance the patient–physician relationship and create social value [12,21]. In the previous study of online health communities, scholars introduced the classic theories of management [22], economics [23], sociology [17,24], and other disciplines [9,25]. The application of interdisciplinary theory further enriches the research of online health communities. As we all know, continuous usage is the basis of utility maximization in online health communities. However, Nijland N et al. [26,27] found that users are declining to use a web-based application for support over time in a mixed-methods way. The analysis of consumer behavior and determinants influencing their decision to continue using is usually performed using some of the well-known traditional theories; for example, Devendra et al. explored that received support is a positive and significant predictor of new users’ continued engagement in OHCs [28]. Wu B found that perceived value and patient satisfaction influence patients’ continued use, integrating the information system success model and online health community features [29]. Zhang et al. clarified that the factors that affect the relationship commitment and continuous knowledge-sharing intention in OHCs integrated social exchange and commitment-trust theories [30]. Liu and Wang identified the factors that influence the continued use of online mental health communities based on the theory of health self-efficacy and expectation confirmation [31]. These studies focus more on individuals and less on the external environment. In order to make a more comprehensive study of the continuous use of online health, we introduced social cognitive theory (SCT) and theories of planned behavior (TPB), which include personal (internal) and environmental (external) factors.

SCT, as one of the most common models, was proposed by Davis and incorporated in a distinctive way how individuals accumulate and retain certain behaviors. It is well established as a powerful and robust model for predicting user acceptance. TPB as a predictor of intentions and behavior has been applied to many studies that contain three variables (i.e., perceived behavioral control, attitude, and subjective norms), which have never been used in OHCs. SCT and TPB are suitable for our research because they systematically explain and predict users’ behavior from personal and environmental perspectives.

### 2.2. Social Cognitive Theory

The social cognitive theory proposes that personal cognitions, behavior, and environment are three variables that dynamically interact with each other [32]. It gives us a better way to know how individual behavior is shaped or changed. Over the past few decades, SCT has been widely used to explain different individual behaviors. Miyao Manabu et al. [33] explored the influences of open innovation hubs and perceived collective efficacy on individual behavior. Mahdizadeh Seyed Mousa et al. [34] applied it to predicting determinants of nosocomial infection control behaviors in hospital nursing staff. Junjie Zhou et al. [35] discussed the factors influencing patient e-health literacy in online health communities. Social cognitive theory is often accepted in depth for examining the reasons why individuals adopt certain behaviors and have proven advantageous for understanding individuals’ behavior in the virtual community [36,37]. The theory contains personal cognitions, behavior, and environmental factors. Personal cognitions include self-efficacy and outcome expectations, which reflect a user’s perceived ability to perform certain activities and his/her expectations toward future outcomes; environment comprised of both physical environment (i.e., system and information) and cultural environment. System quality refers to the user’s perception of the overall performance of a website, such as the degree of access speed, ease of use, visual appeal, and navigation [38,39]. Information quality refers to the user’s perception of the quality of information presented on a website, such as being relevant, sufficient, accurate, and up-to-date [40]. Social interaction ties can be treated as a symbol of the cultural environment, which refers to the strength of the relationships between different interaction parties, the amount of time spent, and the communication frequency among different OHCs members. They act as channels for information and resource flows [41].

### 2.3. Theory of Planned Behavior

The theory of planned behavior (TPB) is the development of the theory of reasoned action (TRA), which aims to explain and predict human behavior and provide a framework for designing behavior change [42]. Fishbein and Ajzen [43] stated that the attitude toward the act and social acceptance of the act could predict behavior toward an act. Later the theory was revised by Ajzen [42] by adding perceived behavior control of individuals on the act to determine the behavioral intention. The theory of planned behavior explains and predicts people’s behavioral decisions mainly through psychological structure, such as attitude and intention, and has generated a range of contributions related to consumer behavior research. TPB has become one of the important theories of human behavior research. According to TPB, behavioral intention has been reported to be the most important construct in deciding whether or not to perform a particular behavior, which in turn can be determined by the attitude toward behavior, subjective norm, and perceived behavioral control [27]. However, despite the fact that TPB received great attention and is still extensively adopted in the prediction of IT usage and intention [44], the theory has rarely been applied in the context of continuance IT usage [45,46] and, more specifically, within the context of this research. In this study, we employ the three predictors of TPB in our research. Perceived behavioral control refers to one’s perceived control in performing a behavior [27], and attitude refers to a person’s favorable or unfavorable evaluation of an object [43,47]. Subjective norms represent social influences on an individual’s engagement in a particular behavior [43].

TPB was well explained in predicting human intentions and users’ behavior, which has been used in marketing domains and information systems. Continuous usage of OHCs means users seek information, obtain help, and other behavior. Meanwhile, OHCs are the products of the development of the Internet, which still belong to the information system. Thus, we introduce this theory in our research, which is also one of the implications of this study.

This paper will attempt to check the factors from both models, SCT and TPB, which can influence the users’ continuous usage in the online health community.

## 3. Research Hypotheses

This paper intends to explore the factors influencing users’ continuous usage in online communities from the TPB and SCT by establishing the research model. Six hypotheses are composed.

### Relationship between Perceived Behavior Control and Continuous Use Intention

Perceived behavioral control refers to an individual’s perception of the amount of control over carrying out the behavior. It is closely related to the perception of how easy or difficult to perform the behavior [27], which contains self-efficacy and controllability [48]. If users do not have adequate control over the behavior, there is not much reason to perform it [49]. Indeed, the literature has empirically proven that perceived behavioral control plays a significant role in driving behavioral intentions [50,51]. Self-efficacy refers to “people’s belief they have the ability to control their own level of functioning and over events that affect their lives” [52]. If users are confident in seeking information and getting support, they will want to continue to use the OHCs.

Conversely, users with low self-efficacy tend to have a negative attitude due to their lack of confidence in their ability to act. Controllability refers to beliefs about the extent to which performing the behavior is up to the actor [48]. If users can easily obtain information in OHCs, they will continue to use the community when they face the same situation. As such, we suggested the following hypotheses:

**Hypothesis** **1a** **(H1a)**.*Self-efficacy has a positive influence on users’ continuous intention in OHCS*.

**Hypothesis** **1b** **(H1b)**.*Controllability has a positive influence on users’ continuous intention in OHCS*.

Attitude refers to the degree to which an individual favors or does not favor the behavior [42]. Research has shown that attitude positively influences and predicts behavioral intention [53,54,55]. As a typical information system, some characteristics of the OHCs are similar to other Internet products. Attitude usually reflects users’ long-lasting preference that an individual has over something. If people have a positive attitude toward participating in OHCs, then people have the intention and will carry out participating activities. In this study, attitude relates to an individual’s judgment concerning the continuous usage of OHCs. Therefore, we suggested the following hypothesis:

**Hypothesis** **2** **(H2)**.*Attitude has a positive influence on users’ continuous intention in OHCs*.

Subjective norms represent social influences on an individual’s engagement in a particular behavior [9]. It is concerned with how the certain behavior of an individual is influenced by the desire to act according to how important referents think she/he should act or as they act themselves [51,56]. In other words, subjective norms are the social pressure from others for an individual to comply with. Previous research has found that subjective norm positively influences behavioral intentions [50,57,58,59]. Nowadays, people are more cautious and emphasize their health issues, so they are more willing to obtain others’ opinions about the health issue. People who they think are important will possibly impact their behavioral intentions and behavior. On this basis, the following hypothesis is proposed:

**Hypothesis** **3** **(H3)**.*Subjective norms have a positive influence on users’ continuous intention in OHCs*.

Members’ intentions can be defined as their specific purpose in acting, the end or goal they aim to accomplish [59]. According to the theory of reasoned action (TRA), intention is assumed to be an indication of an individual’s willingness and readiness to behave [43]. Ajzen [42] has proved that intention to act and realized action has a strong correlation. Lin Judy [60] confirms that the web user’s willingness is a strong predictor of his/her action. Continuous intention is a precondition for continuous usage. When people have the intention to continue usage, they will translate into action if other conditions are already (i.e., controllability and self-efficacy). Therefore, we suggested the following hypothesis:

**Hypothesis** **4** **(H4)**.*Continuous intention has a positive influence on continuous usage in OHCs*.

According to the SCT, individual behavior can be affected by personal cognition and environment [49]. Personal cognition is the core of the theory, which contains self-efficiency and outcome expectation. The SCT factors considered in this study are self-efficacy, outcome expectations, and environmental factors. The definition of self-efficacy is the same as before. It refers to people’s belief that they have the ability to control their own level of functioning and over events that affect their lives [52]. Previous research has found that self-efficacy is a significant determinant of user behavior. When users perceive high self-efficacy, they feel they have the ability and expertise to obtain medical information and help others in OHCs. High self-efficacy may facilitate their continued usage. Otherwise, they may discontinue participating in OHCs. Outcome expectations refer to anticipatory judgments about the likely consequences of one’s actions [35]. When users participate in the OHCs, they can obtain what they want to satisfy their health needs; they will continue using the OHCs. Otherwise, they may discontinue their usage. Researchers have found that outcome expectation positively affects online patient-provider communication. Meng-Hsiang Hsu et al. [46] found that outcome expectations have a positive effect on knowledge-sharing behavior. Therefore, we suggested the following hypothesis:

**Hypothesis** **5a** **(H5a)**.*Self-efficacy has a positive influence on users’ continuous usage of OHCs*.

**Hypothesis** **5b** **(H5b)**.*Outcome expectations have a positive influence on users’ continuous usage of OHCs*.

The environment is an important factor in the theory of behavior. The SCT considers a social environment in which a person performs his/her behavior, which can be influenced by an individual’s characteristics and emotional states [61]. The environment differs from social norms in that it emphasizes individual emotions and characteristics. In contrast, social norms emphasize external pressure from peers, friends, and relatives on individuals. Combined with previous studies [35,41,62] and our research background, the environment in this research contains system quality, information quality, and social interaction ties. System quality and information quality are physical environments, while social interaction ties are the cultural environment. System quality reflects the reliability, ease of use, and visual appeal of community platforms [63], and information quality is used to measure the information system’s output [64]. The information is more truthful, relevant, and helpful. The higher quality the member perceives, the more likely it is that the member will adopt this information [65]. Social interaction ties reflect the relationship strength and interaction frequency between members of a community [41]. The system quality and information quality are essential for users to continue participating in OHCs. If the health information they obtain is accurate, useful, and high quality, they may feel satisfied and continue their usage. The OHCs members interact and communicate with others, forming a health-related activity that acts as an important part of the community sphere. Stronger social interaction ties are powerful indicators of an active community sphere and harmonious relationships [41], which can promote users’ continue participating in OHCs; thus, we suggested the following hypotheses:

**Hypothesis** **6a** **(H6a)**.*System quality has a positive influence on users’ continuous usage of OHCs*.

**Hypothesis** **6b** **(H6b)**.*Information quality has a positive influence on users’ continuous usage of OHCs*.

**Hypothesis** **6c** **(H6c)**.*Social interaction ties have a positive influence on users’ continuous usage of OHCs*.

Based on the literature review and hypothesis, we established a theoretical model shown in Figure 1.

## 4. Methods

### 4.1. Instrument Development

To guarantee the reliability and validity of the questionnaire, we adopted the previous maturity scale with a 7-point Likert-type response format that ranged from strongly disagree to strongly agree. Given that the survey would be conducted in China and our participants were Chinese, the study team conducted a translation process for cross-cultural adaptation. The scale items were translated into Chinese and then translated back into English by different researchers who have held at least a master’s degree and were suitable at speaking English and scientific research translation dependently to ensure the accuracy of the translation. To improve comprehensibility and readability, several participants who had experience in OHCs and had different backgrounds in terms of age, sex, and education level were recruited to complete the questionnaire and provide modifications. In the context of OHCs, the items of the self-efficiency [66], controllability [66], attitude [66,67], and subject norms [66,68] were assessed by Ya-Yueh Shih, outcome expectation [41], and environmental factors (system quality, information quality [69,70] social interaction ties [41] of SCT) were examined by Chiu, C.-M., and Bongsik Shin, the continuous intention [54,71] and behavior were measured by Bhattacherjee A., [72,73]. The specific scale is shown in Appendix A.

### 4.2. Analysis Tool Selection

Structural equation modeling (SEM) is useful in analyzing the causal relationships of research models and accommodating intricate causal networks, and it can help incorporate measurement errors and detect effects. This study adopted partial least squares structural equation modeling to test the hypotheses. Smart PLS version 3.0 was used for the effective and unbiased analysis and assessment of potential variable interactions.

### 4.3. Data Collection and Respondent Profile

The formal investigation was anonymously conducted through a web-based questionnaire survey addressed to participants in December 2021. We sent a web-based questionnaire to 510 participants to create and maintain the questionnaire by Credamo, which is a professional survey and experiment platform. Then, the participants filled out this questionnaire through the platform. The questionnaire automatically terminates when the user chooses to last use the online health community for more than 3 months.

Our participants were individuals who had engaged in OHCs within the previous three months before the questionnaire. A total of 510 responses were received, 480 of which were valid. (The response that was not completed or that missed at least 1 answer is invalid) The validity rates were 94.1% (480/510). Table 1 shows the demographics of the sample. Most of the participants were aged 20–40 years (81.1%; 389/480), 53.3% (256/480) of the participants were female, and 86.5% (415/480) held at least a bachelor’s degree. These figures are not only generally consistent with statistical reports on the development of the Internet in China [74] but also consistent with research that showed that the OHC characteristics were young [75], female, and highly educated. Thus, the scale met the requirement.

### 4.4. Common Method Bias and Non-Response Bias Test

As this study uses single data source data for hypothesis testing, there may be the influence of common method bias. We adopt a Harman single factor to test common method bias. It was found that a total of 4 factors with characteristic roots more significant than one were extracted, and the variation explained by the first factor was 32.17%, less than 40%. Thus, this indicates that this study has no common method bias.

According to the method proposed by Armstrong et al. [76], all potential variables were first and last collected in this study for the non-response bias test. The chi-square test was applied to analyze whether significant differences existed between the earliest 25% and the last 25% of data collected in all measurement items. The results showed they did not differ significantly (*p* > 0.05). Therefore, there is no non-response bias problem in the survey data collected in this study.

## 5. Results

### 5.1. Reliability and Validity

Reliability and validity is the key to data analysis; the data is meaningless if the key indicators are not up to standard. The consistency coefficient Cronbach’s alpha, which has a cut-off value of 0.7, and composite reliability (CR), which has a cut-off value of 0.7, were used to measure the sample reliability. The results show that Cronbach’s alpha and composite reliability of each variable is more significant than 0.8, indicating that the measurement scale in this study has suitable internal consistency and sample reliability. (See Table 2).

Validity is divided into content validity, convergent validity, and discriminant validity. Since all the studies in this paper refer to the maturity scale developed by scholars, they have suitable content validity. For convergent validity, we followed the criteria proposed by Fornell and Larcker [77]. As shown in Table 2, the factor load of each measurement item is higher than the cut-off value of 0.7, and all the AVE values exceeded the cut-off value of 0.500, indicating that the convergent validity of the scales was acceptable. Table 3 shows the results of the discriminant validity. For each construct, the square root of AVE (minimum value is 0.838) was above each correlation between the other construct and itself (maximum value is 0.731). Therefore, the discriminant validity was acceptable.

Before testing the hypothesis, a collinearity test of variables was conducted in this study, which is measured by a variance inflation factor (VIF), according to Hair and Risher’s research. The variance inflation factor of each potential variable in the collinearity study should be less than 5. As shown in Table 2, the VIF of all constructs is less than 4, indicating no collinearity problem in this study, which supports hypothesis testing in the next step.

### 5.2. Hypothesis Testing

After fully verifying the questionnaire’s reliability, validity, and collinearity, we assessed the structural model. Following previous literature, we examined the main effects and assessed the moderation effect.

The structural model analysis is mainly evaluated from three aspects [77]: standardized residual root mean square (SRMR), coefficient of determination (R^2^ value), and significance level of path coefficient. The SRMR value less than 0.08 indicate that the model has a suitable overall fit, and the SRMR estimated in this study is 0.053, which meets the requirements of model fit. The R^2^ value is mainly used to explain the variance of endogenous potential variables, which should be large enough. Table 4 shows the multivariate coefficient of determination (R^2^) with and without control variables; this study used Cohen’s *f*^2^ to assess the effects of control variables. Depending on the criterion of R^2^ (i.e., insignificant: <0.020; small: 0.020 and <0.1 medium: >0.150 and <0.300; large: >0.350). The R^2^ meets the requirements of model fit, and the effect of control variables had limited effects on the constructs.

Table 5 and Figure 2 present the magnitude and significance of the path coefficients, respectively. These results indicate an acceptable level of explanatory power. All hypotheses were supported. In addition, controllability, attitude, self-efficacy, and subjective norms affect continuous intention, which, in turn, affects continuous usage; self-efficacy, outcome expectations, system quality, information quality, and social interaction ties also affect continuous usage.

At last, this study analyzed the mediating effects by using the bootstrapping method (n = 5000, 95% confidence interval (CI)). Confidence intervals generated by bootstrapping were used as the criteria to check whether the indirect effects were significantly different from zero. The mediating effects are significant if zero is not in the confidence intervals [78]. According to Table 6, the indirect effects were significant for the CIs excluded zero. Thus, the mediating effects were significant.

The study results of the path analysis indicated that all path coefficients were significant. Our study data statistically supported our model, and all study hypotheses were accepted with a 0.05 level of significance. The results indicate that self-efficacy and controllability have significant effects on users’ continuous intentions. Thus, H1a and H1b are supported. Attitude has a significant relationship with continuous intention; therefore, H2 is accepted. Moreover, social norms have a positive effect on continuous intention; hence, H3 is supported. Likewise, the relationship between intention and behavior is positive; thus, H4 is supported. Self-efficacy and outcome expectations have significant positive associations with continuous usage; therefore, H5a and H5b are accepted. Finally, significant and positive relationships were found between environmental factors (system quality, information quality, and social interaction tie) and continuous usage; thus, H6a; H6b; H6c are accepted. Therefore, a hypothesized model is acceptable in this study. What is more, self-efficacy, controllability, attitude, and social norms had positive and significant effects on users’ continuous usage through the mediating factors of continuous intention.

## 6. Discussion

### 6.1. Principal Findings

This study is the first to introduce the theory of TPB into users’ continuous usage in OHCs, combined with the theory of SCT; it makes theoretical contributions and practical implications for future study. First, we constructed a research model to explore the factors that affect users’ continuous usage through personal and environmental perspectives. Previous studies have centered on only one theory, such as self-determination theory, social support theory, etc. This study enriches theories and gives some suggestions for the development of OHCs in China.

Second, the result of this study showed that perceived behavior control (self-efficacy and controllability), attitude, and social norms have significant effects on the positive impacts of continuous intention. The results are similar to TPB in other areas. Zeynep Erden et al. [59] have proven that attitude, subjective norms, and perceived behavior control can affect users’ intention to share knowledge in online communities. This further broadens the application of TPB. As a result, among the factors, the effects of social norms on the continuous intention in OHCs is the more significant than other factors. This may be because people are more affected when accepting new things. They are inclined to obtain suggestions from people whom they think are important or authority, for they think the authorities are highly professional; for example, when the COVID-2019 epidemic broke out, the Chinese people were more willing to trust academician Zhong Nanshan and follow his instructions. Another explanation is that we are a collectivistic society. Although someone can benefit from the online health community, they may discontinuously use it because there is no one around them to use it. Online health community suppliers can invite authorities to provide suggestions and guide people to obtain medical information more efficiently.

Third, the result in this study is consistent with previous studies showing that both personal cognitions (outcome expectation) and environmental factors (system quality, information quality) have significant effects on continuance behavior. Among them, information quality has the largest effect, similar to that of Zhang et al. [79], who found the effect of information quality on users’ loyalty toward brand micro-blogs. As users access online communities for valuable information, it is reasonable to conclude that information quality is a crucial factor determining their behavior, especially in online health communities, for users visit it with a strong purpose compared with other online communities; they need to obtain health-related information. If the information quality is poor, users cannot find the utility and value of participating in a community, and they will discontinue their engagement. Furthermore, it is also necessary to provide users with a reliable and easy-to-use platform. What is more, the results of this study show that self-efficacy and social interaction ties have positive effects on continuous usage. This is partly consistent with previous studies [41,80] and partly differs from previous research [62]. Combined with our research, the explanation is that users could make a friend or create social interaction ties that may be weaker than family ties, but it still supplies ways for users to communicate, obtain information and support, and then forms a special social circle. Some users who are individualistic may think self-efficacy is critical for continuous usage, and they obtain what they want when needed in OHCs. Others treasure the social ties in which OHCs will continually participate.

### 6.2. Theoretical and Practical Implications

From a theoretical perspective, this research draws on the theory of planned behavior, which has never been used in OHCs as the typical theory of behavior, to examine users’ continuous usage in OHCs. Although previous research has examined online health community user behavior from multiple perspectives, such as the theory of social support and the motivational theory, TPB, which has been widely used and validated for human behavior in numerous contexts, has never been applied in OHCs. This study is the first to introduce it into users’ continuous usage in OHCs, enrich extant research, and advance our understanding of online health community users’ behavior.

The second finding of this research is that applied TPB and SCT mixed model to investigate factors that influence users’ behavior, which is more comprehensive than SCT by subdividing personal and environmental factors. Cognition means personal inner cognition, self-efficacy, and outcome expectation, which come from users’ toward themselves, and external perception comes from users’ toward external objects (controllability). Meanwhile, environmental factors contain not only online factors (system quality, information quality) but also other factors, for example, some important people’s opinions and users’ attitudes.

From a practical perspective, this paper has two practical implications for users’ continuous usage. First, online health service providers must consider personal cognitions and environmental factors to facilitate and stimulate user continuance. Otherwise, the community cannot survive in the marketplace. On the one hand, they need to try to change people’s cognition gradually, for some people still think that the effect of online health consultation is not as suitable as traditional medical. Providers should let people know online consultation has lots of advantages, such as getting services from some reputable doctor without a crowd and giving users more discounts for continuous usage and other incentive policy. On the other hand, a quality platform may help improve users’ satisfaction and facilitate their continuance. They need to organize the website’s page layout, navigation menus, and site elements to ensure users easily search and find what they want. They also need to ensure a quality community (system quality and information quality) offered to users and improve users’ experience. Then, users may be satisfied and continue their usage.

The other contribution is to decisions and policymakers. The Chinese government has introduced a series of policies to promote the development of Internet medical services since 2018 [81], such as “The Opinions on Promoting the Development of “Internet plus Medical Health,” “Guidance from National Healthcare Security Administration on improving the price of “Internet plus” medical service the price of “Internet plus” medical services insurance payment policies” [82,83]. Firstly, they should provide more favorable policies to stimulate and help providers build and manage strategic plans for sustaining a thriving online community; for example, provide the same rate of payment for users and encourage traditional medical institutions to transfer some business to OHCs, especially during the COVID-19 pandemic in recent years. Secondly, they should build a supportive climate in society through public service advertisements and invite authorities to provide popularization lectures. In addition, developing a national Internet infrastructure and the medical system’s resources will relieve the pressure of traditional medical treatment while achieving the ”Healthy China 2030” strategy in the long run.

## 7. Limitations

Even though this study has offered valuable insights into users’ continued usage, it has some limitations, as most field surveys suffered. Firstly, the study did not consider community type. A variety of professional online health communities were explored in this study. However, it is not clear that the community means doctor to patient or patient to patient or other types. In addition, traditional hospitals are also adjusting their business to online platforms. Future researchers could do more accurate study then provide more clear and specific suggestions. Another limitation is that the environmental factors in SCT theory are not comprehensive enough, which only contain the online factors. Future studies are encouraged to consider offline environments such as habit and other constructs to deepen insight. The last limitation of this study is that the dates were obtained from China, which neglects the different cultural content; this may affect the study’s conclusion. Future studies should consider this in the research.

## 8. Conclusions

This study applied TPB and SCT mixed model to investigate the users’ continuous usage of OHCs by subdividing personal and environmental factors. Personal factors include self-efficacy, outcome expectation, and attitude; environmental factors mainly refer to the external perception of the individual; on the one hand, they come from the feelings brought by external influential people, which are the subjective norms in the study; the other is the individual’s feelings when they are participating in the online health community in OHCs, such as information quality, system quality, and social interaction ties in this research.

The results confirm that both personal and environment have positive effects on users’ intentions and behavior. Therefore, the platform’s management should focus on changing users’ cognition by cultivating a positive environment to erode users’ concerns. Some people still think that the effect of online health consultation is not as suitable as traditional medical meanwhile, there is a risk of user privacy disclosure. Providers should gradually let people know that online consultation has lots of advantages, such as the fact that some people who come from poor counties can obtain services from a reputable doctor without a crowd. Users can choose doctors who specialize in their field. Furthermore, they can put forward more beneficial measures for users to continue to use, such as giving users more discounts, establishing a membership model, and other incentive measures. In addition, the community can protect users’ personal information from being leaked through advanced science and technology and strict confidentiality codes.

The information quality affects users’ behavior most among all the factors in this research. Thus, managers should continuously improve the information and system quality to help users obtain what they want. To improve the quality of information, the platform should review the quality of doctors’ comments and set administrators to manage the information, comments, and advertisements effectively in the community. Big data and artificial intelligence are suitable ways for the patient to achieve accurate calculations. With the coming of the aging society, simple operation and being easy to use is the basis of improving the use of the elderly; system stability is critical when users are participating in OHCs; intelligent speech recognition and personalized service are also supplements for users to quickly find relevant information. All above provide directions for the supplier to improve the quality of the community system.

Policy support is also crucial for OHCs’ development. The government could provide more medical subsidies to patients after they use online health services and provide the OHCs operator some preferential tax. The traditional hospital can provide doctors more job recognition and performance rewards when they carry out online diagnosis and treatment services, which gradually shunt some patients online and promote the progress of hospitals’ online business in the long run.

This research provided theoretical guidance for community platform operators to stipulate effective measures to encourage users’ continuous usage, offering a reference for decisions and policymakers.

## Figures and Tables

**Figure 1 healthcare-11-01238-f001:**
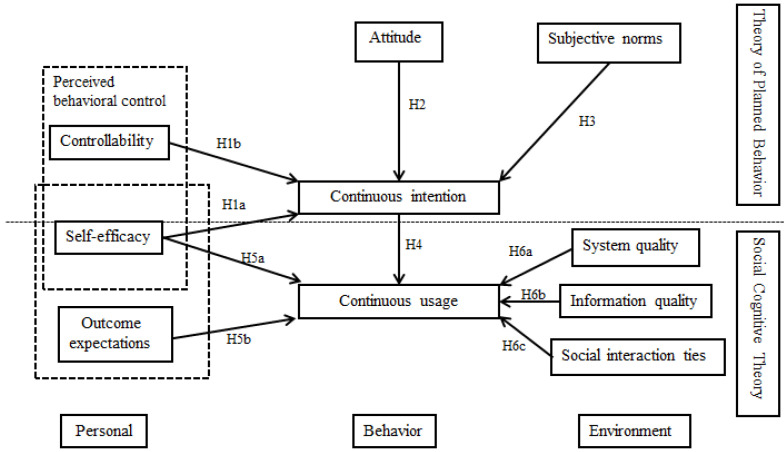
Research model.

**Figure 2 healthcare-11-01238-f002:**
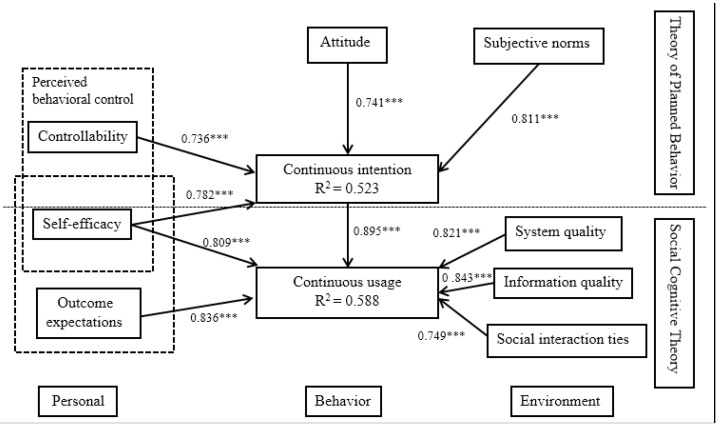
Research model with path coefficients. *** *p* < 0.001.

**Table 1 healthcare-11-01238-t001:** Sample demographics (n = 480).

Demographic Characteristics	Participants, n(%)
Age (years)	
<20	17 (3.5)
20–29	222 (46.3)
30–39	167 (34.8)
40–49	61 (12.7)
50–59	10 (2.1)
60 and above	3 (0.6)
Gender	
Male	224 (46.7)
Female	256 (53.3)
Living area	
Urban	310 (64.6)
Rural	170 (35.4)
Education	
Junior hihg school and below	15 (3.1)
High school	50 (10.4)
Bachelor’ degree	209 (43.5)
Master’ degree	173 (36.1)
Doctor’s degree and above	33 (6.9)

**Table 2 healthcare-11-01238-t002:** Cronbach alpha of constructs, CR, AVE, VIF, and square root of AVE.

Constructs	Item	Standard Factor Loading	Variance Inflation Factor	Cronbach’s Alpha	CR	AVE	Square Root of AVE
Self-efficacy	S1	0.93	1.437	0.916	0.893	0.702	0.838
	S2	0.89					
	S3	0.91					
Controllability	C1	0.96	1.533	0.932	0.926	0.732	0.856
	C2	0.91					
	C3	0.93					
Attitude	A1	0.91	1.426	0.907	0.836	0.721	0.849
	A2	0.89					
	A3	0.93					
Subjective norms	SN1	0.96	1.506	0.918	0.917	0.717	0.847
	SN2	0.86					
	SN3	0.91					
Continuous intention	CI1	0.92	1.455	0.924	0.913	0.785	0.886
	CI2	0.94					
	CI3	0.89					
Continuous usage	CB1	0.93	1.328	0.915	0.907	0.752	0.867
	CB2	0.91					
	CB3	0.91					
Outcome expectations	OE1	0.91	1.314	0.910	0.914	0.783	0.885
	OE2	0.92					
	OE3	0.88					
System quality	SQ1	0.88	1.306	0.905	0.903	0.724	0.851
	SQ2	0.89					
	SQ3	0.92					
	SQ4	0.91					
Information quality	IQ1	0.95	1.369	0.922	0.924	0.741	0.861
	IQ2	0.93					
	IQ3	0.84					
	IQ4	0.89					
Social interaction ties	ST1	0.92	1.411	0.916	0.886	0.794	0.891
	ST2	0.93					
	ST3	0.89					
	ST4	0.91					

**Table 3 healthcare-11-01238-t003:** Fornell–Larcker test of the discriminant validity.

Variable	SE	C	SN	CI	CU	OE	SQ	IQ	SIT
SE	0.838								
C	0.529	0.856							
A	0.473	0.533							
SN	0.556	0.654	0.847						
CI	0.689	0.387	0.652	0.886					
CU	0.731	0.571	0.630	0.651	0.867				
OE	0.322	0.388	0.407	0.496	0.611	0.885			
SQ	0.546	0.476	0.606	0.569	0.357	0.521	0.851		
IQ	0.433	0.519	0.618	0.393	0.408	0.637	0.593	0.861	
SIT	0.497	0.475	0.425	0.544	0.511	0.706	0.436	0.675	0.891

**Table 4 healthcare-11-01238-t004:** Multivariate coefficient of determination (R^2^) results.

Variables	R Square		Control Variables Effects		
	With Control Variables	Without Control Variables	ΔR² ^a^	*f* ^2 b^	Effect
CI	0.523	0.521	0.002	0.015	Insignificant
CU	0.588	0.583	0.005	0.008	Insignificant

^a^ ΔR^2^: R^2^ with control variables-R^2^ without control variables; ^b^
*f*: Cohen *f*^2^.

**Table 5 healthcare-11-01238-t005:** Hypothesis testing.

Hypothesis	Path Coefficients	*t* Test	*p* Value
H1a	0.736	14.31	<0.001
H1b	0.782	14.66	<0.001
H2	0.741	14.24	<0.001
H3	0.811	16.56	<0.001
H4	0.895	17.79	<0.001
H5a	0.809	16.31	<0.001
H5b	0.836	16.95	<0.001
H6a	0.821	16.73	<0.001
H6b	0.843	17.32	<0.001
H6c	0.749	14.57	<0.001

**Table 6 healthcare-11-01238-t006:** Path coefficients by bootstrapping.

Hypothesis	Path Coefficients	*p* Value	CI
**Direct effect**			
SE → CI	0.736	<0.001	0.631–0.824
Controllability → CI	0.782	<0.001	0.675–0.879
Attitude → CI	0.741	<0.001	0.639–0.813
SN → CI	0.811	<0.001	0.719–0.903
CI → CU	0.895	<0.001	0.787–0.985
SE → CU	0.809	<0.001	0.723–0.856
OE → CU	0.836	<0.001	0.748–0.889
SQ → CU	0.821	<0.001	0.736–0.917
IQ → CU	0.843	<0.001	0.756–0.836
SIT → CU	0.749	<0.001	0.685–0.793
Indirect effect			
SE → CU	0.563	<0.001	0.489–0.612
Controllability → CU	0.498	<0.001	0.399–0.528
Attitude → CU	0.577	<0.001	0.496–0.654
SN → CU	0.374	<0.001	0.306–0.452
Total effect			
SE → CU	0.563	<0.001	0.489–0.612
Controllability → CU	0.498	<0.001	0.399–0.528
Attitude → CU	0.577	<0.001	0.496–0.654
SN → CU	0.374	<0.001	0.306–0.452

## Data Availability

The data presented in this study are available from the corresponding author upon reasonable request.

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
