# Peer review of "Factors Affecting Users’ Continuous Usage in Online Health Communities: An Integrated Framework of SCT and TPB"

_healthcare, 2023, doi:10.3390/healthcare11091238_

Round 1

Reviewer 1 Report

A detailed review was attached.

Author Response

Dear reviewers

Re:Manuscript ID: healthcare-2314628 and Title: Factors Affecting Users' continuous usage in Online Health Communities: An Integrated Framework of SCT and TPB

Thank you for taking time and effort to go through our paper and providing constructive comments which are valuable and helpful for us. We have read through comments carefully and provided a point-by-point response to your comments. Please see the attachment.

Best wishes

Sincerely.

Reviewer 2 Report

I believe this study is timely.

I felt your clearly described your methodology and results very clearly.   

I appreciate your recognition of the potential limitations of this study.   

I would like to have seen the discussion and conclusions expanded a little more.   I think that the conclusions specifically could have been expanded upon.

Author Response

Dear reviewers

Re:Manuscript ID: healthcare-2314628 and Title: Factors Affecting Users' continuous usage in Online Health Communities: An Integrated Framework of SCT and TPB

We are grateful to you for taking time and effort to go through our paper and give your suggestions which are valuable and helpful for us. We have read through suggestions carefully and have provided a point-by-point response to your suggestion. Please see the attachment.

Best regards

Sincerely.

Reviewer 3 Report

Dear Editors

Dear authors,

Thank you for the opportunity to read and review this paper. I really enjoy to read this manuscript about users’ continuous usage is of great value for the long-term development of OHCs.

Before considering publishing this paper, this paper needs several improvements.

Abstract section

Please add the purpose of this study.

What is the theoretical framework used in this study?

How online questionnaires and SEM can answer the research questions?

Literature review is well written.

It would be good if authors can provide figures or link or framework or diagram about online health community. This information will be useful for readers.

Methods section

Methods section must be rewritten. Please use this framework for write methods section.

1.      sample and participants ( who is the participants of this study? please add Data demographics respondent)

2.      data collections (how the author collected the data?)

3.      questionnaires (how authors develop the questionnaires?

4.      data analysis (how authors analysis and interpreted the data?)

5.      what software has been used to analysis the data? AMOS? SMART-PLS?

6.      please explain about CB-SEM.

results section

please add normality test result

please provide HTMT for discriminant validity. fornell-lacker test not accurate.

Please added the explanation of cross loading, SRMR, NFI, RMS-Theta

what is the value of R2 in this study? explain the explanatory power of structural model

what is the value of indirect effect? please added.

Theoretical and practical implications is well written.

Author Response

 Dear reviewers

Re:Manuscript ID: healthcare-2314628 and Title: Factors Affecting Users' continuous usage in Online Health Communities: An Integrated Framework of SCT and TPB

We are grateful to you for taking time and effort to go through our paper and give your comments which are valuable and helpful for us. We have read through comments carefully and have made changes. Especially methods section has rewritten which may not follow your framework exactly mainly because we're running out of time. However, the current framework is still reasonable. Please see the attachment.

Best regards

Sincerely.

Round 2

Reviewer 1 Report

It can be accepted in present form.

Author Response

Dear reviewers

Re:Manuscript ID: healthcare-2314628 and Title: Factors Affecting Users' continuous usage in Online Health Communities: An Integrated Framework of SCT and TPB

It is our great honor to get your recognition for this research, and thank you again for your constructive comments on this paper.

Best regards

Sincerely.

Zhuolin Cao

Reviewer 3 Report

Dear Editor,

Dear authors,

this article have been revised well.

article can be accepted after minor revision.

Reference must be improved. How many references can proof how deep author understand and master this topic.

Best

Author Response

Dear reviewers

Re:Manuscript ID: healthcare-2314628 and Title: Factors Affecting Users' continuous usage in Online Health Communities: An Integrated Framework of SCT and TPB

Thank you again for taking time and effort to go through our paper and providing constructive comments which are valuable and helpful for us. We have added some literature, particularly in the area of online health community research which has been a little bit neglected. Please see the new version.

Best wishes

Sincerely.

Zhuolin Cao